# Interband Electron Transitions Energy in Multiple HgCdTe Quantum Wells at Room Temperature



**Nikolay N. Mikhailov** [1], **Sergey A. Dvoretsky** [1,*], **Vladimir G. Remesnik** [1], **Ivan N. Uzhakov** [1], **Vasyliy A. Shvets** [1] and **Vladimir Ya. Aleshkin** [2]

[1]   A.V. Rzhanov Institute of Semiconductor Physics of Siberian Branch of Russian Academy of Sciences, Novosibirsk 630090, Russia

[2]   Institute for Physics of Microstructures of RAS, Nizhny Novgorod 603087, Russia

[*]   Correspondence: dvor@isp.nsc.ru

**Abstract:** The studies of the interband electron transition energy in multiple $Hg_{1-x}Cd_xTe/Hg_{1-y}Cd_yTe$ quantum wells (MQWs) at room temperature were carried out. The MQWs were grown on the (013) GaAs substrate by molecular beam epitaxy, with the layer composition and thickness being measured by the in-situ ellipsometric parameters measurement at the nanometer level. The $Hg_{1-x}Cd_xTe$ barrier composition and width were x = 0.69 and 30 nm, respectively. The $Hg_{1-y}Cd_yTe$ well composition was y = 0.06–0.10, and the width varied in the range of 2.7–13 nm. The experimental data of the interband electron transition energy were determined by the absorption spectral analysis. The calculation of the interband electron transition energy was carried out on the basis of the four-band Kane model. A good agreement between the experimental and calculated data was obtained. It was shown that MQWs may be used as a photosensitive material for creating infrared optoelectronic devices operating in different modes in the range of 3–10 μm at room temperature.

**Keywords:** energy; interband transition; HgCdTe; multiple quantum wells; absorption

## 1. Introduction

A solid mercury and cadmium telluride (MCT, HgCdTe) solution, due to its unique physical properties, such as the band gap change in a wide energy range, the large absorption coefficient, and the high carrier mobility and lifetime, has taken the leading position among the photosensitive materials for high-quality infrared (IR) detectors [1–3]. The numerical research and development of epitaxial technologies such as LPE, MOVPE, and MBE have made it possible to obtain high-quality HgCdTe layers and cooled IR detectors in the wide spectral range of 1 to 20 μm.

However, the growing process needs the development of MOVPE and MBE growth on large-diameter, low-cost GaAs and Si substrates [4–6] instead of high-cost CdZnTe substrates [4,7]. Now IR detectors based on HgCdTe/GaAs(Si) have parameters that are comparable to the similar ones based on HgCdTe/CdZnTe in the 1–3 μm and 3–5 μm ranges at 77 K. However, for a long IR detector wavelength over 10 μm, it is necessary to cool HgCdTe/GaAs(Si) IR detectors to temperatures lower than 50 K. The fundamental requirement of large area HgCdTe layers for long IR detector wavelength is composition uniformity over the substrate area. This problem becomes practically unsolvable with the increase in the long wavelength cutoff.

The quantum nanostructures with layer thicknesses from several to tens of nanometers, such as HgTe/CdTe superlattices (SLs) and HgTe/HgCdTe quantum wells (QWs), are considered alternative materials for creating IR detectors that can be used in a wide spectral range at elevated temperatures [8–11].

It was shown that the IR detector sensitivity on the basis of HgTe/CdTe SLs is determined by the thickness relationship of the CdTe and HgTe layers [12]. That can essentially

---

increase the wavelength cutoff uniformity over a large area. The first HgTe/CdTe SLs were grown by MBE because this method is most preferable due to the low growth temperature [13]. Nevertheless, there are no positive results in using HgTe/CdTe SLs for developing high-quality IR detectors, in spite of the significant progress in the HgCdTe alloy growth technology. This may be connected with the absence of control over the layer thickness accuracy and their stability during the CdTe/HgTe SL growth. The CdTe and HgTe thickness relation determines the interband electron transition energy (IBETE) and, as a result, the IR detector spectral range sensitivity. A small layer width change in the very narrow CdTe and HgTe layers may lead to IBETE blurring and an essential change in the spectral characteristics, magneto-transport, and magneto-optical phenomena [14–16].

The spectral sensitivity of multiple quantum wells (MQWs) of $Hg_{1-x}Cd_xTe/Hg_{1-y}Cd_yTe$ is determined only by the $Hg_{1-y}Cd_yTe$ well composition y and width ($d_w$) at the large barrier composition x and width ($d_b$). The electron interband transition in $Hg_{1-x}Cd_xTe/Hg_{1-y}Cd_yTe$ MQWs allows one to realize a highly sensitive IR detector [10]. For this, it is necessary to grow high-quality $Hg_{1-x}Cd_xTe/Hg_{1-y}Cd_yTe$ MQWs with very precise layer control composition and width. The MBE technique is the preferred method to satisfy such a requirement. We developed the MBE technology of the HgCdTe alloy, different periods of $Hg_{1-x}Cd_xTe/Hg_{1-y}Cd_yTe$ QWs growth, and single-wave ellipsometry (SWE) monitoring of layer composition and thickness in situ. The SWE has obviously the following advantages: high speed and precision, which are very important for growing a nanometer-thick layer; they do not affect growth processes; and the interpretation of measurements can be carried out in real time during growth without the interactive participation of an operator.

This work presents a complex study of the $Hg_{1-x}Cd_xTe/Hg_{1-y}Cd_yTe$ MQWs via the MBE growth and absorption spectra. The MQWs were grown by MBE on (013) GaAs substrates with ellipsometric control in situ to provide high precision of the layer composition and width repetition. The absorption spectra were extracted from the optical measurements of the transition and reflection spectra. The analyzed data of the MQWs parameters allowed calculating the band structures of MQWs of different well widths. The experimental and calculated interband electron transitions in $Hg_{1-x}Cd_xTe/Hg_{1-y}Cd_yTe$ MQWs depended on the $Hg_{1-y}Cd_yTe$ well composition and width. The obtained data confirm the possibility of fabricating optoelectronic IR devices in a wide spectral range at room temperature.

## 2. Materials and Methods

The $Hg_{1-x}Cd_xTe/Hg_{1-y}Cd_yTe$ MQWs were grown by MBE on (013)CdTe/ZnTe/GaAs alternative substrates with ZnTe (50 nm thick) and CdTe (5–7 µm thick) buffer layers sequentially in the multi-chamber UHV MBE set "Ob-M," without removal to the atmosphere [17]. The special design of Te, Cd, and Hg molecular beam sources and their arrangement relative to the substrate in the technological growth chambers provides HgCdTe composition uniformity over the layer surface no worse than 0.0002 cm$^{-1}$ along 4 inches in diameter *without* substrate rotation [18]. All technological chambers are equipped with stable, high-speed, and very precise laser ellipsometers (λ = 632.8 nm) [19,20], which allow monitoring the growth process in situ [21]. The composition and thickness measurement accuracy are Δx(y) = 0.0005 and Δd = 0.5 nm, respectively.

The HgCdTe layer composition is determined by the optical constants n and k. The optical constant data at typical HgCdTe growth of 180–190 °C were obtained from the measurement of the ellipsometric parameters ψ and Δ of thick layers with different compositions and are described by the following expression [22]:

$$n(x,y) = 3.967 - 0.92(x,y) \tag{1}$$

$$k(x,y) = 1.327 - 2.819(x,y) + 4.432(x,y)^2 - 4.375(x,y)^3 + 1.7(x,y)^4 \tag{2}$$

The HgCdTe layer thickness at MQW growth is determined from the ψ and Δ measurements in ψ-Δ plane, as described in [23]. The behavior of ψ and Δ during the growth of CdTe/HgTe SL was theoretically predicted in [24].

The determination of the composition distribution throughout the MQW layer thickness was carried out by dividing the entire thickness into fractions of nanometer fragments (FNF) with a subsequent calculation of their optical constants by solving the inverse ellipsometry problem [25] using the developed method of the "effective" substrate [26]. The highly stable laser ellipsometer [24] provides the random errors of the ellipsometric parameters $\delta\Psi \approx 0.003°$ and $\delta\Delta \approx 0.01°$. At the typical fragment layer thickness of 0.3 nm, the composition measurement accuracy is $\Delta y = 0.01$ and $\Delta x \sim 0.03$ near the QW bottom and top, respectively.

The well widths ($d_w$) and numbers of periods (N) of the investigated $Hg_{1-x}Cd_xTe/Hg_{1-y}Cd_yTe$ MQWs are presented in Table 1. The barrier composition and width were $x = 0.62$ and 30 nm, respectively.

**Table 1.** The $Hg_{0.38}Cd_{0.62}Te/Hg_{1-y}Cd_yTe$ well width under the growth and study.

| Sample, No | #1 | #2 | #3 | #4 | #5 | #6 | #7 | #8 | #9 | #10 | #11 | #12 | #13 | #14 | #15 | #16 | #17 | #18 |
|---|---|---|---|---|---|---|---|---|---|---|---|---|---|---|---|---|---|---|
| $d_w$, nm | 2.7 | 3.2 | 3.6 | 4.0 | 4.4 | 4.4 | 5.4 | 5.9 | 6.1 | 6.9 | 7.4 | 7.8 | 8.1 | 8.4 | 9.4 | 11.0 | 12.3 | 12.8 |
| N, un. | 10 | 10 | 5 | 5 | 5 | 5 | 5 | 5 | 5 | 5 | 5 | 5 | 5 | 5 | 10 | 10 | 10 | 10 |

The IBETE were determined by analysis of the absorption (A) spectra. The experimental A values were extracted from the transmission (T) and reflection (R) spectra using the relation $A = \log(1/(T + R))$. The T and R spectra were measured at 300 K using the FTIR spectrometer "FT-801" (Simex Ltd., Novosibirsk, Russia) equipped with a special attachment in the wave number range 470–8500 $cm^{-1}$ (21 μm–1.2 μm) with an accuracy of not worse than 0.5 $cm^{-1}$.

The calculation of the IBETE was provided on the basis of a developed four-band Kane model ($8 \times 8$ Hamiltonian) taking into account deformation effects. An explicit form of the Hamiltonian for a structure grown on the (013) plane is given in [27]. To solve the Schrödinger equation, an approach similar to that proposed in [28] was used, but not polynomials, but the Fourier series [27,29] were used as the basis for the expansion of the wave function. The essence of the method is that the electronic states are calculated not in a structure with a single quantum well but in a superlattice with wide tunnel-nontransparent barriers. The advantage of this approach is that the inconvenient boundary condition for the wave function to vanish at infinity for a structure with a single quantum well is replaced by a convenient calculation condition for the wave function periodicity over the superlattice period. The band gap dependence on the composition of the solid solution and temperature was taken from [1], and other Kane Hamiltonian parameters were taken from [30]. To describe the deformation contribution to the Hamiltonian, the parameters were taken from ref. [31]. Due to the small spin splitting, the subbands are arranged in pairs.

## 3. Results

### 3.1. MQWs Growth

It is clear that 100 MQW periods can ensure the fabrication of highly sensitive IR detectors [10]. To implement this, it is necessary to ensure the following over time: optimal growth conditions when very thin layers with large differences in composition x and y must have a good crystallinity; high reproducibility of the barrier and well-defined composition x and y and their width. Such requirements were successfully realized by our numerous basic studies of the HgCdTe alloy growth on the (013) GaAs substrate together with the development of the high speed and accuracy SWE control in the UHV MBE unit without substrate rotation.

In Figure 1, the typical dependence of the ellipsometric parameters $\Delta$ and $\psi$ variation on time for the growth of 40-period MQW HgTe/$Hg_{0.3}Cd_{0.7}Te$ is shown. At the initial stage, 50 nm thick $Hg_{0.3}Cd_{0.7}Te$ layers were grown on the (013)CdTe surface of the alternative substrate CdTe/ZnTe/GaAs (see Section 2) before the beginning of growing 40 period HgTe/$Hg_{0.3}Cd_{0.7}Te$ MQW with 5.2 nm well width and 8 nm barrier width, respectively.

Then the large variation of Δ and ψ is observed during the growth of the first 10 periods. After this, Δ and ψ parameter variations decrease and practically disappear after the growth of the 20th period. In spite of behavior variations at the growth and parameter levels, the composition and thickness of each HgTe and $Hg_{0.3}Cd_{0.7}Te$ layer in their pairs are constant. The analogous Δ and ψ variations must be observed at the nanostructure's growth based on HgCdTe or any matched materials with large differences in optical constants.

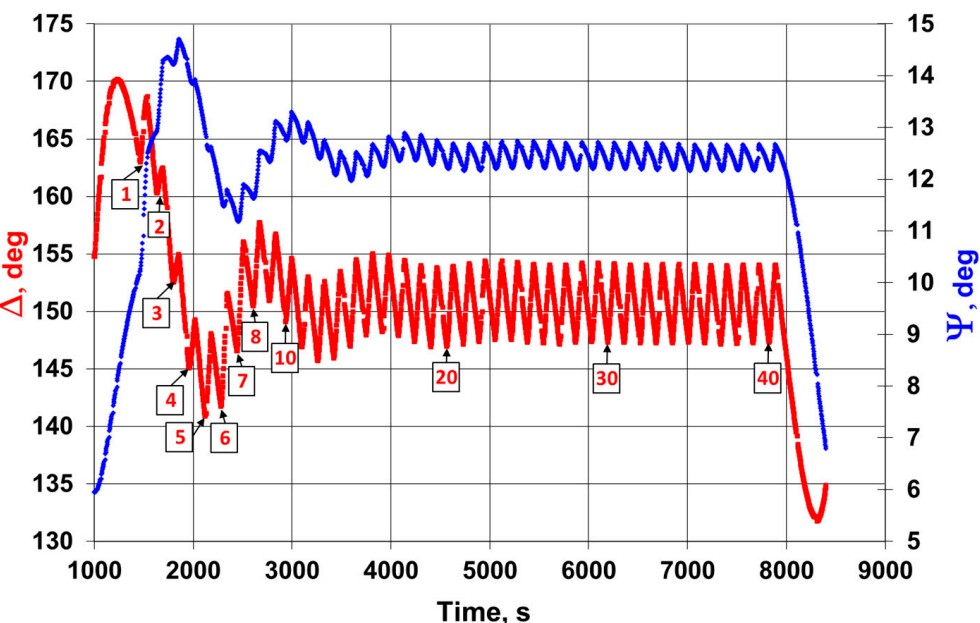

**Figure 1.** The change of the ellipsometric parameter Δ (red) and ψ (blue) measured in situ vs. time during 40 periods of the MQW growth. The numbers show the period of the barrier and well pair growth.

A detailed composition distribution throughout the thickness in the first 20 periods of the $HgTe/Hg_{0.3}Cd_{0.7}Te$ MQW is shown in Figure 2.

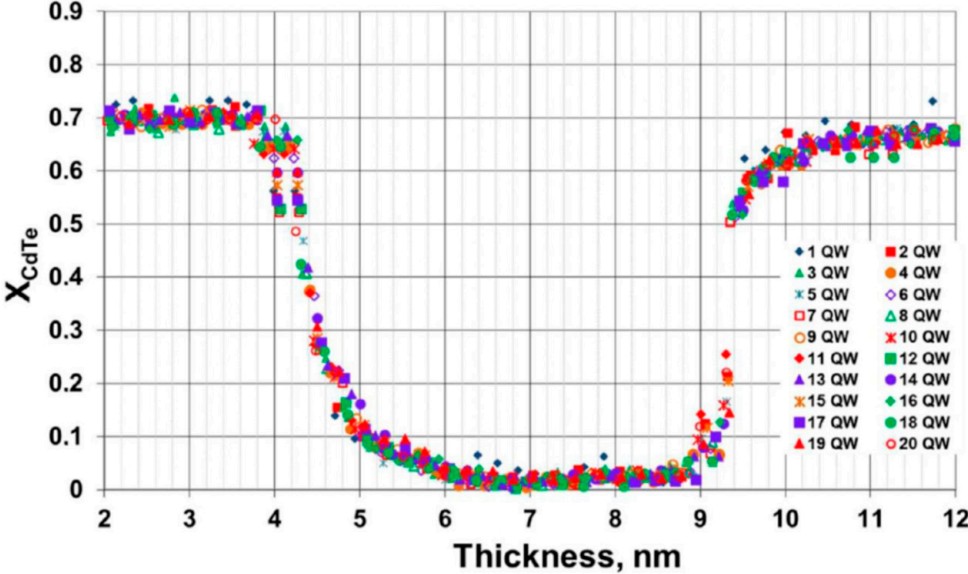

**Figure 2.** The composition distribution throughout the thickness for the initial 20 periods of the $HgTe/Hg_{0.3}Cd_{0.7}Te$ MQW. The color symbols represent the well number.

The composition variations were restored for 20 initial periods in the MQW by using the method of the "effective" substrate on the basis of the experimental ellipsometric data $\Delta$ and $\psi$ measured at the growth. Excellent repetition and composition variation throughout the thickness are seen. This means that the growth conditions were very stable and the ellipsometric parameters were measured with high precision.

As a result, the MQW growth technology is very reproducible. At the HgTe growth between $Hg_{0.3}Cd_{0.7}Te$ barriers, the composition is decreased from y = 0.7 to y = 0.0 at the thickness of 2 nm, becomes constant at the thickness of 3 nm, and is increased at the thickness of 1 nm to x = 0.7. We evaluated the quantum well thickness of 5.2 nm from the close to open cadmium molecular beam. Such composition distribution throughout the thickness was determined by the specificity of the technology process and an error when restoring the composition. The last reason is connected with the relative error of ellipsometric measurements and the FNF thickness. The accuracy of composition recovery is greater with increasing FNF thickness, whose determination accuracy decreases, and vice versa: if the spatial resolution is increased by decreasing the FNF thickness, this will lead to an increase in the scatter of the determined composition. In our case, the typical FNF thickness is 0.3 nm, and the composition measurement accuracy is $\Delta y = 0.01$ and $\Delta x \sim 0.03$ near the QW bottom and top, respectively.

Additionally, to determine the IBETE, 5–10 period $Hg_{1-x}Cd_xTe/Hg_{1-y}Cd_yTe$ MQWs were grown (see Table 1). The barrier provided a tunnel-like, opaque layer between wells. The well width was determined by the closing and opening of the Cd molecular beam source shutter. In Figure 3, the composition distribution in 5 periods of MQWs with $d_w = 3.6$ nm (a) and $d_w = 8.1$ nm (b) is shown.

After closing the shutter, the composition in the wells sharply decreases from y = 0.62 to y = 0.2 and then slightly to y = 0.05. When the shutter opens, the well composition sharply increases to y = 0.6. Such results attest to the good composition distribution throughout the well's width. The observed composition change was determined by the specific Cd molecular beam source operation that allows supporting a constant molecular flux of not worse than 1% over a long time and to provide a large number of MQW growth processes.

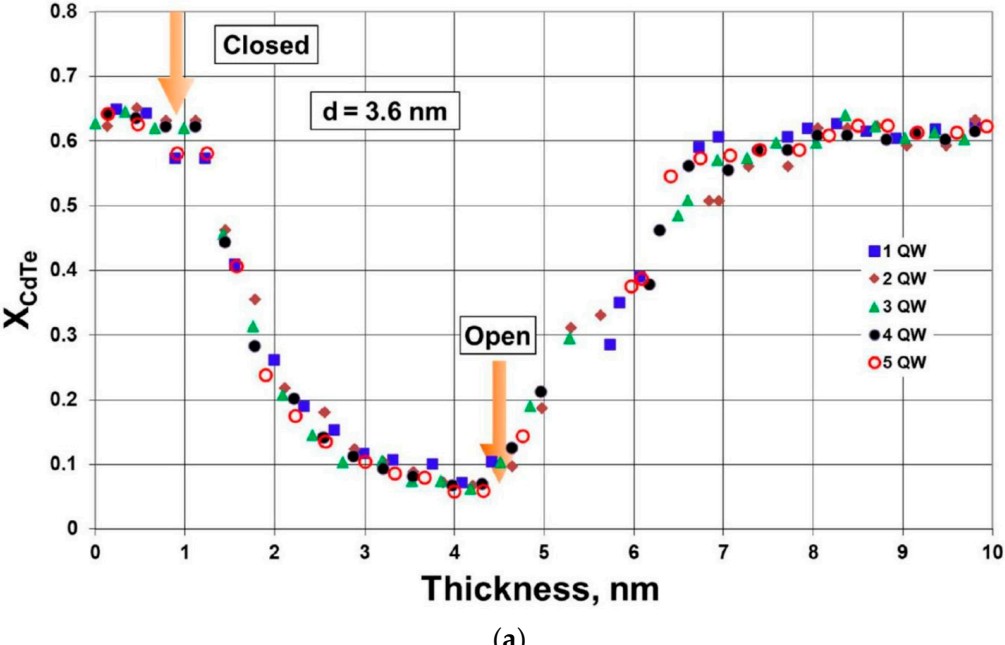

(a)

**Figure 3.** *Cont.*

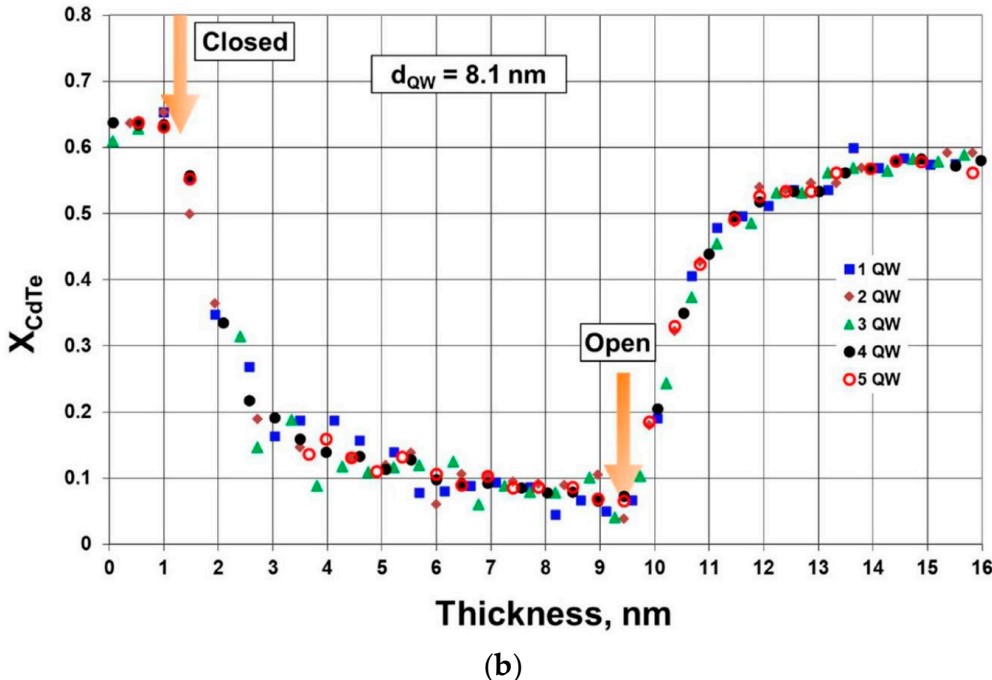

**(b)**

**Figure 3.** The composition distribution throughout the thickness of a 5-period MQW, with different wells represented by different color points: (**a**) $d_w$ = 3.6 nm; (**b**) $d_w$ = 8.1 nm.

### 3.2. Band Structure

In Figure 4, the calculated band structure of two samples (#3 (a) and # 13 (b)) for room temperature is shown, and the cadmium content was obtained by averaging the composition distribution throughout the well width shown in Figure 3. The calculation showed that MQWs #3 and #13 are semiconductors with normal band structures. It can be seen from this data that the band structure for sample #3 has one electron band (red curve), three heavy hole subbands HH(1–3), and a light hole LH1. For sample #13, the band structure is more complex. There are two electron subbands, E1 and E2, six heavy spin-orbit splittings, HH1–6 (only five shown for room temperature), and one light hole subband.

Table 2 shows the energy of the allowed electronic interband transition between subbands made at room temperature.

In the samples, the minimal IBETE for the fundamental E1-HH1 (values highlighted in red) is decreasing with the well width. It is necessary to note the change of heavy and light hole subbands in the energy position and their rearrangement with the change in well width. In sample #3, the LH1 energy is higher than that of HH2. In sample #13, the LH1 energy is lower than that of HH2. It is clear that the band structure for samples with some Cd content in the well changes with the well width approaching the critical value, but more slowly than in the case of pure HgTe. The fundamental E1-HH1 IBETE determines the wavelength cutoff for the IR detector or the laser wavelength radiation emission, which corresponds to 4.35 μm and 7.75 μm, respectively.

**Table 2.** The energy of the allowed electronic transition between the conduction and valence subbands for different samples #3 and #13 at room temperature. The fundamental electronic transition energy for samples #3 and #13 is represented by the red color.

| Sample No | IBETE, meV | | | | | |
|---|---|---|---|---|---|---|
| | **E1-HH1** | **E1-LH1** | **E1-HH3** | **E1-HH5** | **E2-HH2** | **E2-HH4** |
| #3 | 285 | 363 | 468 | - | - | - |
| #13 | 160 | 212 | 248 | 359 | 412 | 525 |

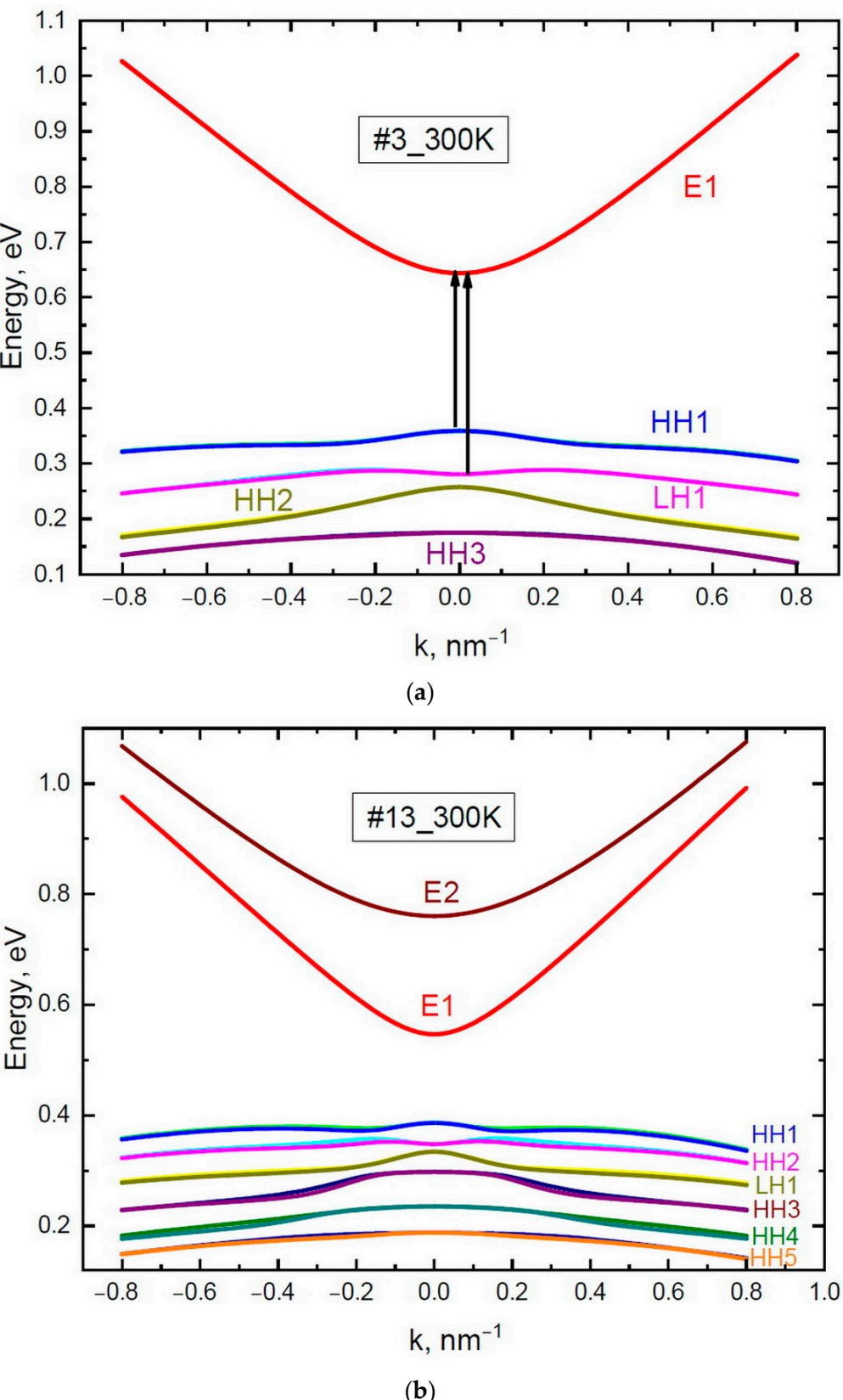

**Figure 4.** Band structure $Hg_{1-x}Cd_xTe/Hg_{1-y}Cd_yTe$ QW with different well width values: (**a**) sample #3 (well width 3.7 nm); (**b**) sample #13 (well width 8.1 nm). E1–red; E2–brown; HH1–blue; HH2–pink for #3 and yellow for #13; HH3–brown; HH4–green; HH5–orange; LH1–yellow for #3 and pink for #13. The arrows represent the allowed electronic transition.

### 3.3. Absorption Spectra

The absorption spectra demonstrate quite a few steps at the absorption edges between the conduction and valence subbands. The absorption edge is defined as the energy at the maximum value of the first derivative of the absorption. In Figure 4, the absorption spectra and their first derivatives for samples #2 (a) and #15 (b) at room temperature are shown. The parasitic maxima in the absorption curves are due to interference effects that are observed in the transmission and reflection spectra.

The first derivative curve showed two maxima (electronic transitions E1-HH1 and E1-LH1) for sample #2 (Figure 5a) and four maxima (electronic transitions E1-HH1, E1-LH1, E2-H2, and E2-H4) for sample #15 (Figure 5b), which corresponded to the absorption of photons with different energies. It is clear that we can reveal more IBETE when the well width increases because of the decrease in the energy gap between the conductance and valence subbands. The full width at a half maximum is 40–50 meV and is determined by the temperature.

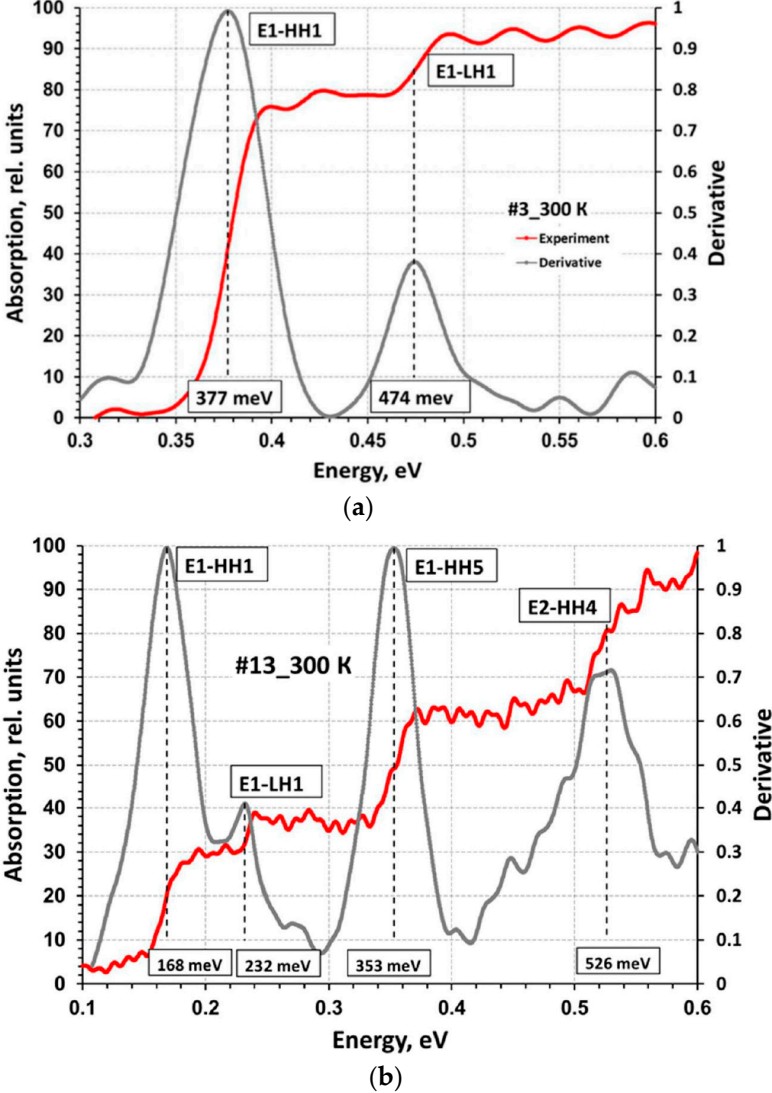

**Figure 5.** Experimental absorption (red lines) and their first derivatives (grey lines) for the samples #2 and #15 at room temperature. The interband electronic transition energy is indicated by vertical lines. In squares are shown E1-HH1, E1-LH1 for the sample #2 and E1-HH1, E1-LH1, E2-HH2, E2-HH4 for the sample #15 interband electronic transition and their energies.

## 4. Discussion

The band structure of the $Hg_{1-x}Cd_xTe/Hg_{1-y}Cd_yTe$ QW depends on barrier (x) and well (y) composition and their widths [32,33]. When the HgTe (y = 0) thickness increases, the QW band structure changes from a normal to an inverted one with a zero-band gap at the critical thickness of 6.3 nm [34,35]. The critical thickness increases with the increasing Cd composition in the well (y) and reaches 8 nm and 17 nm at y = 0.05 and y = 0.1 [36]. It means that all the grown MQWs in this study are semiconductors with a normal band structure. It was shown that SLs with a normal band structure are the more promising material for long-wavelength and very long-wavelength IR detectors [12]. So, we can conclude that the MQWs with the Cd composition y = 0.1–0.15 in the well are a normal-band semiconductor for developing different photonic devices at wide infrared and terahertz wavelength ranges and temperatures. The barrier width of 30 nm in MQWs was enough to exclude the interaction between carriers in neighboring wells and ensure the fundamental IBETE was independent of the well width, as shown for the Sls in [9]. The precisely developed repetition of barriers and well parameters during the large periods of MQW growth on the (013) orientation can ensure the same band structure in each well and, thereby, a good quantum efficiency and sensitivity.

In Figure 6, the IBETEs dependences on well width in $Hg_{1-x}Cd_xTe/Hg_{1-y}Cd_yTe$ MQWs are extracted from the band structure calculation and experimental absorption spectra.

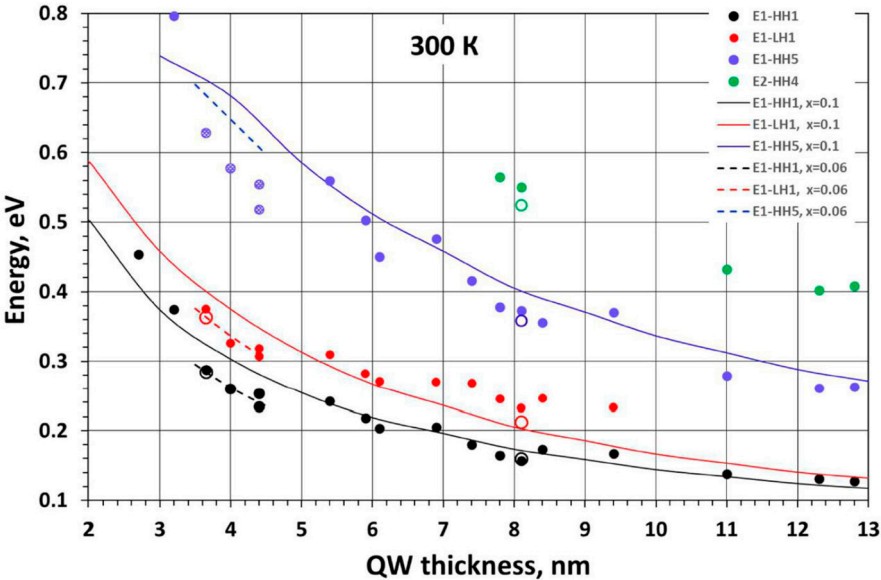

**Figure 6.** IBETE's dependences on well width in MQWs. The solid and open circles are the experimental and calculated IBETEs values. The solid and dashed curves are the calculated data for MQWs with constant Cd composition x = 0.1 and x = 0.06: black curve–E1-HH1; red curve–E1-LH1, blue curve–E2-HH2. The green solid circles are for E2-HH4 IBETEs.

It is seen that IBETE's values decrease with increasing well thickness. The experimental and calculated IBETEs data for the investigated MQWs are in good agreement with the calculated data for ideal MQWs with rectangular walls obtained by fitting with the help of well composition as a parameter. Most of the investigated MQWs have a composition distribution throughout their thickness that is the same as in ideal MQWs with the constant well composition x = 0.1 (solid curves). The four MQWs have a different composition distribution throughout the thickness that is the same as in ideal MQWs with constant well composition x = 0.06 (dashed curves). The change in composition distribution throughout the well thickness is determined by different Cd molecular beam source control operations. That allows MQWs to grow with the required IBETEs and, thereby, the fundamental edge, which determines the basic wavelengths of devices. The green circles, which corresponded to IBETEs, determine the MQWs with a well thickness of more than 8 nm.

In Table 3, the experimental and calculated fundamental E1-HH1 IBETEs values for investigated MQWs and the proposed device wavelength ($\lambda$) at 300 K are collected. There is a good agreement between experimental and calculated data for fundamental interband electron transition.

**Table 3.** Fundamental IBETE E1-HH1 and wavelength values for the proposed devices.

| Sample, No | #1 | #2 | #3 | #4 | #5 | #6 | #7 | #8 | #9 | #10 | #11 | #12 | #13 | #14 | #15 | #16 | #17 | #18 |
|---|---|---|---|---|---|---|---|---|---|---|---|---|---|---|---|---|---|---|
| $d_w$, nm | 2.7 | 3.2 | 3.7 | 4.0 | 4.4 | 4.4 | 5.4 | 5.9 | 6.1 | 6.9 | 7.4 | 7.8 | 8.1 | 8.4 | 9.4 | 11.0 | 12.3 | 12.8 |
| E1-HH1exp., meV | 454 | 375 | 288 | 260 | 253 | 242 | 234 | 218 | 203 | 205 | 180 | 173 | 168 | 163 | 157 | 138 | 131 | 128 |
| E1-HH1calc., meV | 480 | 403 | 285 | 253 | 266 | - | 180 | 163 | 233 | 163 | 177 | 116 | 160 | 168 | 131 | 105 | 75 | 105 |
| $\lambda$exp., $\mu$m | 2.73 | 3.31 | 4.31 | 4.77 | 4.90 | 5.12 | 5.30 | 5.69 | 6.11 | 6.05 | 6.89 | 7.17 | 7.43 | 8.00 | 7.90 | 8.99 | 9.47 | 9.69 |

As shown for HgTe (4.15 nm)/Hg$_{0.05}$Cd$_{0.95}$Te (8.95 nm) SL, which have a normal band structure, the E1-HH1 IBETE at room temperature is ~180–200 meV [14]. This value is lower than for our MQW (sample #4) with the same well width. Such a difference is determined by the presence or absence of Cd content in wells. In our case, the Cd content of the well is x = 0.06. For the HgTe/Hg$_{0.05}$Cd$_{0.95}$Te SL, there is no Cd content in wells. It is clear from our results (see Figure 6) that the IBETEs for our MQWs decrease with the decreasing Cd content in wells and will reach the analogous values in the absence of Cd in wells (i.e., pure HgTe) as for the HgTe/Hg$_{0.05}$Cd$_{0.95}$Te SL at room temperature.

The Hg$_{1-x}$Cd$_x$Te/Hg$_{1-y}$Cd$_y$Te MQWs with the normal band structure are a good choice for developing different IR photonic devices. The detector sensitivity and laser efficiency will be determined by the fundamental E1-L1 IBETE values. It is clear that Hg$_{1-x}$Cd$_x$Te/Hg$_{1-y}$Cd$_y$Te MQWs with x > 0.6 and y < 0.1 may be used for developing electronic devices in the wavelength range 3–10 $\mu$m at room temperature and provide precise control of the required spectral range by changing the Cd content and well width.

The results of the practical use of Hg$_{1-x}$Cd$_x$Te/Hg$_{1-y}$Cd$_y$Te MQWs observation of the optically pumped stimulated emission (SE) in the 2.5–3.0 $\mu$m wavelength range at room temperature were published in [37].

## 5. Conclusions

The interband electron transition energy and its dependence on the well width and Cd content in multiple semiconducting Hg$_{1-x}$Cd$_x$Te/Hg$_{1-y}$Cd$_y$Te quantum wells were investigated at room temperature.

The MQW growth with different well periods on the (013) GaAs substrate by MBE with the precise ultra-speed ellipsometric control in situ was demonstrated. The high repeatability of the Cd composition distribution throughout the well thickness in 40-period MQWs was shown.

The experimental IBETE positions were extracted from the absorption spectra obtained from the measurement of the transition and reflection spectra of the 5–10 period MQWs, with the well width in the range of 3–13 nm.

The theoretical model for IBETE positions of the MQW's on (013) orientations for the real Cd profile was developed. The calculated IBETE's for MQWs are in good agreement with the experimental data for the MQWs with a constant Cd content in wells. It was found that the experimental IBETE values depended on the Cd content of MQWs.

The MQWs with the Cd content in wells have a normal band structure in the wider range of well width, different from that for pure HgTe. That allows fabricating material with planned IBETE positions, especially for the fundamental E1-H1 transition. The variation of the fundamental E1-H1 transition in the range 450–130 meV in MQWs with well widths of 3–13 $\mu$m will be used to determine the operation wavelength of the optoelectronic devices in the range 3–10 $\mu$m at room temperature.

**Author Contributions:** Conceptualization, S.A.D., N.N.M. and V.Y.A.; methodology, V.G.R., V.A.S. and I.N.U.; investigation, N.N.M., V.G.R. and V.Y.A.; writing-original draft preparation, S.A.D., N.N.M. and V.Y.A.; writing review and editing, S.A.D.; visualization, N.N.M. and V.Y.A.; supervision, S.A.D. All authors have read and agreed to the published version of the manuscript.

**Funding:** The work was partially supported by the Russian Foundation for Fundamental Research (project No. 21-52-12015) and within the framework of the Program of Fundamental Scientific Research of the State Academies of Sciences, topic code FWGW-2022-0002.

**Institutional Review Board Statement:** Not applicable.

**Informed Consent Statement:** Not applicable.

**Data Availability Statement:** The data presented in this study are available on request from the corresponding author.

**Conflicts of Interest:** The authors declare no conflict of interest. The funders had no role in the design of the study, in the collection, analysis, or interpretation of data, in the writing of the manuscript, or in the decision to publish the results.

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
