# Peer review of "Interband Electron Transitions Energy in Multiple HgCdTe Quantum Wells at Room Temperature"

_photonics, doi:10.3390/photonics10040430_

Round 1

Reviewer 1 Report

In this manuscript, the interband electron transition energy in semiconducting multiple Hg1-xCdxTe/Hg1-yCdyTe quantum wells were investigated at room temperature by the adsorption spectral analysisas well as the theory calculation base on the four band Kane model. A good agreement between the experimental and calculated data was obtained. However, there are some problems, which must be solved before it is considered for publication.

1、 Does the orientation of the substrate have any effect on the Hg1-xCdxTe/Hg1-yCdyTe multiple quantum wells energy band structure, and why did the GaAs substrate choose the (013) direction instead of other crystal orientations?

2、 Since the manuscript emphasizes that the Hg1-xCdxTe/Hg1-yCdyTe multiple quantum wells were grown on the GaAs substrate, in the second paragraph of section 3.1, why was the material grown on the (013) CdTe surface?

3、 In the last paragraph of section 2, the calculation of the IBETE was provided on the basis of a developed four-band Kane model, taking into account deformation effects. How the deformation effect was considered, it is suggested that the authors give an explanation about this, because the deformation factor will affect the final energy band structure.

4、 In the conclusion of section 4, it is mentioned that the MQWs with the Cd composition y=0,1-0,15 in the well are a normal band semiconductor for developing different photonic devices at wide infrared and terahertz wavelength ranges and temperatures. Then how to judge that for a normal energy bands, the Cd component must be in y=0,1-0,15? There is no obvious logical reasoning in the context and the reviewer suggests the authors to give an explanation in detail.

5、 The manuscript needs careful editing and particular attention to English grammar, spelling, and sentence structure. Such as in Abstract, “MWQs”would be “MQWs”, “The MQWs were grown on the (013) GaAs substrate by molecular beam epitaxy with the layer composition and thickness by the ellipsometric parameters measurement in-situ at the nanometer level”would be“The MQWs were grown on the (013) GaAs substrate by molecular beam epitaxy with the layer composition and thickness were measured by the in-situ ellipsometric parameters measurement at the nanometer level, “The Hg1-yCdyTe well composition and width were y=0,06-0,10 and the width varied in the range of 2,7– 13 nm”would be “The Hg1-yCdyTe well composition was y=0,06-0,10 and the width varied in the range of (2,7– 13) nm”etc.

Author Response

Thank you very much for your efforts concerning our manuscript. We also thank the Reviewer for their time spent reviewing the manuscript. We appreciate the positive assessment of our paper.  

1、 Does the orientation of the substrate have any effect on the Hg1-xCdxTe/Hg1-yCdyTe multiple quantum wells energy band structure, and why did the GaAs substrate choose the (013) direction instead of other crystal orientations?

Authors answer:

The symmetry of the structure grown on the (013) plane differs from that of the structure grown on the (001) plane. In addition, shear elements are present in the elements of the strain tensor, see reference [27]. However, this circumstance has little effect on the electronic spectrum in the conduction band, which is practically isotropic. In the valence band, there are differences in the band structure of the samples grown on the (013) and (001) planes.

The orientation of the GaAs (013) substrate was chosen empirically after carrying out numerous experiments to determine the optimal orientation, from our current point of view, for the growth of HgCdTe. The (013) orientation makes it possible to grow high-quality HgCdTe layers with a composition change over a wide range with the required electrical and optical characteristics under changing growth conditions in a wide range of the Hg/Te ratio and temperature compared to the (112) orientation, which made it possible to grow multiple high quality multiple quantum wells. The GaAs substrate was chosen as a wafer of large size (up to 4 inches in diameter), low cost, high mechanical strength compared to CdZnTe substrates.

2、 Since the manuscript emphasizes that the Hg1-xCdxTe/Hg1-yCdyTe multiple quantum wells were grown on the GaAs substrate, in the second paragraph of section 3.1, why was the material grown on the (013) CdTe surface?

Authors answer:

In section 2 the sentence “The Hg1-xCdxTe/Hg1-yCdyTe MQWs were grown by MBE on (013)GaAs substrates with ZnTe (50 nm thick) and CdTe (5-7 μm thick) buffer layers in the multi-chamber UHV MBE set “Ob-M” [17] has been changed to «The Hg1-xCdxTe/Hg1-yCdyTe MQWs were grown by MBE on (013)CdTe/ZnTe/GaAs alternative substrates with ZnTe (50 nm thick) and CdTe (5-7 μm thick) buffer layers sequentially the multi-chamber UHV MBE set “Ob-M”, without removal to the atmosphere [17]».

In section 3.1 the sentence “At the initial stage, 50 nm thick Hg0.3Cd0.7Te layers was grown on the (013)CdTe surface before the beginning of growing 40 period HgTe/Hg0.3Cd0.7Te MQW, 5.2 nm well width and 8 nm barrier widths, respectively.» has been changed to «At the initial stage, 50 nm thick Hg0.3Cd0.7Te layers was grown on the (013)CdTe surface of the alternative substrate CdTe/ZnTe/GaAs (see section 2) before the beginning of growing 40 period HgTe/Hg0.3Cd0.7Te MQW, 5.2 nm well width and 8 nm barrier widths, respectively.»

3、 In the last paragraph of section 2, the calculation of the IBETE was provided on the basis of a developed four-band Kane model, taking into account deformation effects. How the deformation effect was considered, it is suggested that the authors give an explanation about this, because the deformation factor will affect the final energy band structure.

Author answer:

Reference [27] gives an explicit form of the strain tensor and the corresponding part of the Hamiltonian due to strain

4、 In the conclusion of section 4, it is mentioned that the MQWs with the Cd composition y=0,1-0,15 in the well are a normal band semiconductor for developing different photonic devices at wide infrared and terahertz wavelength ranges and temperatures. Then how to judge that for a normal energy band, the Cd component must be in y=0,1-0,15? There is no obvious logical reasoning in the context and the reviewer suggests the authors to give an explanation in detail.

In the second sentence of Section 4, it is indicated that the critical thicknesses of quantum wells with cadmium fractions of 0.05 and 0.1 are 8 and 17 nm. At the critical thickness, the band gap vanishes. When the quantum well thickness is less than the critical one, the band structure is normal, and when the critical thickness is exceeded, the band structure is inverted.

5、 The manuscript needs careful editing and particular attention to English grammar, spelling, and sentence structure. Such as in Abstract, “MWQs”would be “MQWs”, “The MQWs were grown on the (013) GaAs substrate by molecular beam epitaxy with the layer composition and thickness by the ellipsometric parameters measurement in-situ at the nanometer level”would be“The MQWs were grown on the (013) GaAs substrate by molecular beam epitaxy with the layer composition and thickness were measured by the in-situ ellipsometric parameters measurement at the nanometer level, “The Hg1-yCdyTe well composition and width were y=0,06-0,10 and the width varied in the range of 2,7– 13 nm”would be “The Hg1-yCdyTe well composition was y=0,06-0,10 and the width varied in the range of (2,7– 13) nm”etc.

Authors answer:

We carefully checked and corrected the text following the comments and suggestions.

Thank you very much for your efforts concerning our manuscript. We also thank the Reviewer for their time spent reviewing the manuscript. We appreciate the positive assessment of our paper.  

1、 Does the orientation of the substrate have any effect on the Hg1-xCdxTe/Hg1-yCdyTe multiple quantum wells energy band structure, and why did the GaAs substrate choose the (013) direction instead of other crystal orientations?

Authors answer:

The symmetry of the structure grown on the (013) plane differs from that of the structure grown on the (001) plane. In addition, shear elements are present in the elements of the strain tensor, see reference [27]. However, this circumstance has little effect on the electronic spectrum in the conduction band, which is practically isotropic. In the valence band, there are differences in the band structure of the samples grown on the (013) and (001) planes.

The orientation of the GaAs (013) substrate was chosen empirically after carrying out numerous experiments to determine the optimal orientation, from our current point of view, for the growth of HgCdTe. The (013) orientation makes it possible to grow high-quality HgCdTe layers with a composition change over a wide range with the required electrical and optical characteristics under changing growth conditions in a wide range of the Hg/Te ratio and temperature compared to the (112) orientation, which made it possible to grow multiple high quality multiple quantum wells. The GaAs substrate was chosen as a wafer of large size (up to 4 inches in diameter), low cost, high mechanical strength compared to CdZnTe substrates.

2、 Since the manuscript emphasizes that the Hg1-xCdxTe/Hg1-yCdyTe multiple quantum wells were grown on the GaAs substrate, in the second paragraph of section 3.1, why was the material grown on the (013) CdTe surface?

Authors answer:

In section 2 the sentence “The Hg1-xCdxTe/Hg1-yCdyTe MQWs were grown by MBE on (013)GaAs substrates with ZnTe (50 nm thick) and CdTe (5-7 μm thick) buffer layers in the multi-chamber UHV MBE set “Ob-M” [17] has been changed to «The Hg1-xCdxTe/Hg1-yCdyTe MQWs were grown by MBE on (013)CdTe/ZnTe/GaAs alternative substrates with ZnTe (50 nm thick) and CdTe (5-7 μm thick) buffer layers sequentially the multi-chamber UHV MBE set “Ob-M”, without removal to the atmosphere [17]».

In section 3.1 the sentence “At the initial stage, 50 nm thick Hg0.3Cd0.7Te layers was grown on the (013)CdTe surface before the beginning of growing 40 period HgTe/Hg0.3Cd0.7Te MQW, 5.2 nm well width and 8 nm barrier widths, respectively.» has been changed to «At the initial stage, 50 nm thick Hg0.3Cd0.7Te layers was grown on the (013)CdTe surface of the alternative substrate CdTe/ZnTe/GaAs (see section 2) before the beginning of growing 40 period HgTe/Hg0.3Cd0.7Te MQW, 5.2 nm well width and 8 nm barrier widths, respectively.»

3、 In the last paragraph of section 2, the calculation of the IBETE was provided on the basis of a developed four-band Kane model, taking into account deformation effects. How the deformation effect was considered, it is suggested that the authors give an explanation about this, because the deformation factor will affect the final energy band structure.

Author answer:

Reference [27] gives an explicit form of the strain tensor and the corresponding part of the Hamiltonian due to strain

4、 In the conclusion of section 4, it is mentioned that the MQWs with the Cd composition y=0,1-0,15 in the well are a normal band semiconductor for developing different photonic devices at wide infrared and terahertz wavelength ranges and temperatures. Then how to judge that for a normal energy band, the Cd component must be in y=0,1-0,15? There is no obvious logical reasoning in the context and the reviewer suggests the authors to give an explanation in detail.

In the second sentence of Section 4, it is indicated that the critical thicknesses of quantum wells with cadmium fractions of 0.05 and 0.1 are 8 and 17 nm. At the critical thickness, the band gap vanishes. When the quantum well thickness is less than the critical one, the band structure is normal, and when the critical thickness is exceeded, the band structure is inverted.

5、 The manuscript needs careful editing and particular attention to English grammar, spelling, and sentence structure. Such as in Abstract, “MWQs”would be “MQWs”, “The MQWs were grown on the (013) GaAs substrate by molecular beam epitaxy with the layer composition and thickness by the ellipsometric parameters measurement in-situ at the nanometer level”would be“The MQWs were grown on the (013) GaAs substrate by molecular beam epitaxy with the layer composition and thickness were measured by the in-situ ellipsometric parameters measurement at the nanometer level, “The Hg1-yCdyTe well composition and width were y=0,06-0,10 and the width varied in the range of 2,7– 13 nm”would be “The Hg1-yCdyTe well composition was y=0,06-0,10 and the width varied in the range of (2,7– 13) nm”etc.

Authors answer:

We carefully checked and corrected the text following the comments and suggestions.

Reviewer 2 Report

In this paper, the authors give very detailed investigations on the MBE growth technology of Hg1-xCdxTe/Hg1-yCdyTe MQWs on (0 1 3) GaAs substrate. The growth processes are monitored by using in-situ ellipsometric spectrometers. The interband transition energies of different MQWs are calculated and measured, and good agreements between numerical and experimental data are achieved. The paper is helpful for developing infrared optoelectronic devices using MCT.

There are many typos. The authors should revise their manuscript carefully. Some of the minor errors are listed below.

1)      “lifetime of,”-> “lifetime”.

2)      “needs determined”: ” determined” should be deleted.

3)      “thickn the layer”?

4)      “no affect growth”?

5)      “adsorption” -> “absorption”.

6)      “(013)GaAs” -> “(013) GaAs”.

7)      “in in”?

8)      “wave function periodicity the over the superlattice”?

9)      “MWQ” -> “MQW”.

10)   In Figure 1, the captions should be revised.

11)   “In Fig. 2 is shown”: please revise such sentences.

Author Response

Thank you very much for your efforts concerning our manuscript. We also thank the Reviewer for their time spent reviewing the manuscript. We appreciate the positive assessment of our paper.  

There are many typos. The authors should revise their manuscript carefully. Some of the minor errors are listed below.

1)      “lifetime of,”-> “lifetime”.

2)      “needs determined”: ” determined” should be deleted.

3)      “thickn the layer”?

4)      “no affect growth”?

5)      “adsorption” -> “absorption”.

6)      “(013)GaAs” -> “(013) GaAs”.

7)      “in in”?

8)      “wave function periodicity the over the superlattice”?

9)      “MWQ” -> “MQW”.

10)   In Figure 1, the captions should be revised.

11)   “In Fig. 2 is shown”: please revise such sentences.

Authors answer:

We carefully checked and corrected the text following the comments and suggestion.

Reviewer 3 Report

Reviewer’s suggestions:

1.     In abstracts section, the data description in numerical typing format should be checked for example x=0,69.

2.     In Figure 1, the variable parameter in axis should be typed in English.

Author Response

Thank you very much for your efforts concerning our manuscript. We also thank the Reviewer for their time spent reviewing the manuscript. We appreciate the positive assessment of our paper.  

Reviewer’s suggestions:

  1. In abstracts section, the data description in numerical typing format should be checked for example x=0,69.
  2. In Figure 1, the variable parameter in axis should be typed in English.

Authors answer:

We carefully checked and corrected the text following the comments and suggestion.

Round 2

Reviewer 1 Report

The data description in numerical typing format should be checked further, for example "5,2nm" " 0,6" etc. 

Author Response

Thank you very much for your additional efforts in reading our corrected manuscript. We thank you for very careful reviewing the manuscript. We appreciate once more the positive assessment of our paper. 

1、 The data description in numerical typing format should be checked further, for example "5,2nm" " 0,6" etc. 

Authors answer:

We check our manuscript and correct numerical typing format.

Reviewer 2 Report

The authors responsed my comments in an appropriate manner.

Author Response

Dear Reviewer,

Thank you very much for your positive Comments. We check our manuscript once more and correct following of the comment of Reviewer 1  
